# Using sociotechnical theory to understand medication safety work in primary care and prescribers' use of clinical decision support: a qualitative study

Mark Jeffries [1,2] Nde-Eshimuni Salema,[1,3] Libby Laing,[1,3] Azwa Shamsuddin,[4] Aziz Sheikh,[5] Tony Avery,[1,3] Antony Chuter,[3] Justin Waring,[6] Richard Neil Keers[2,7]

For numbered affiliations see end of article.

**Correspondence to**
Dr Mark Jeffries;
mark.jefferies@manchester.ac.uk

## ABSTRACT

**Objectives** The concept of safety work draws attention to the intentional work of ensuring safety within care systems. Clinical decision support (CDS) has been designed to enhance medication safety in primary care by providing decision-making support to prescribers. Sociotechnical theory understands that healthcare settings are complex and dynamically connected systems of fluid networks, human agents, changing relationships and social processes. This study aimed to understand the relationship between safety work and the use of CDS.

**Design and setting** This qualitative study took place across nine different general practices in England. Stakeholders included general practitioners (GPs) and general practice-based pharmacists and nurse prescribers. Semi-structured interviews were conducted to illicit how the system was used by the participants in the context of medication safety work. Data analysis conducted alongside data collection was thematic and drew on socio-technical theory.

**Participants** Twenty-three interviews were conducted with 14 GPs, three nurse prescribers and three practice pharmacists between February 2018 and June 2020.

**Results** Safety work was contextually situated in a complex network of relationships. Three interconnected themes were interpreted from the data: (1) the use of CDS within organisational and social practices and workflows; (2) safety work and the use of CDS within the interplay between prescribers, patients and populations; and (3) the affordances embedded in CDS systems.

**Conclusion** The use of sociotechnical theory here extends current thinking in patient safety particularly in the ways that safety work was co-constituted with the use of CDS alerts. This has implications for implementation and use to ensure that the contexts into which such CDS systems are implemented are taken into account. Understanding how alerts can adapt safety culture will help improve the efficacy of CDS systems, enhance prescribing safety and help to further understand how safety work is achieved in primary care.

## STRENGTHS AND LIMITATIONS OF THIS STUDY

⇒ The use of sociotechnical theory to frame the thematic analysis enabled a nuanced understanding of the ways in which the clinical decision support was utilised for medication safety work.

⇒ This study had a longitudinal design with a variety of stakeholders from different health professions in primary care, interviewed over time across multiple geographical locations.

⇒ Semi-structured interviews were conducted with a full range of different prescribers in different roles (general practitioners (GPs), nurses and pharmacists).

⇒ Many practice staff were drawn from two of the four targeted clinical commissioning groups areas due to recruitment difficulties related to GP time and workload.

⇒ This study relied solely on interview data and may have benefited from observations or other ethnographic approaches that might have uncovered 'work-as-done'.

## INTRODUCTION

Healthcare services are increasingly interpreted as complex systems comprising fluid networks of people and technologies that interact in dynamic, non-linear and interdependent ways.[1–4] New thinking in the field of patient safety, often termed Safety-II, that focuses on 'work-as-done', rather than 'work-as-imagined', understands that safety is achieved less because people follow strict rules but rather because people adapt and respond to unanticipated events.[3 5] This idea departs from the 'measure and manage' orthodoxy of patient safety policies by attending to the situated practices of care work in the context of wider and inherently complex socio-cultural systems.[6 7] The concept of 'safety work' captures the idea that clinicians routinely

need to engage in purposeful activities to actively create and restore safety in the context of dynamic and changeable systems.[8] Safety work has been understood from the concept of the multidimensional and dynamic safety culture of primary healthcare organisations.[9] This offers a way of exploring the interplay between system factors and also the purposeful behaviours of clinicians that goes beyond the reductionist and linear view of established safety perspectives, and instead foregrounds the combined influence of context and work as shaping clinical safety. Safety work has been seen to entail an understanding of safety culture among staff,[9] systems and collaboration between reception, administration and medical staff,[10] articulation work,[11] development of relationships and engagement with stakeholders[12 13] and management of workload.[8]

Sociotechnical theory that draws on interdependent and constructionist theories considers that people and technology are reciprocally and recursively related and as a consequence the outcomes of the relationships between the social, human agency and technology are considered as interdependent and not simply as the interactions between homogeneous unique elements.[14 15] This approach focuses on the social processes involved in the use of the technology since it sees technological use and adoption as a social practice that may involve negotiation and conflict. Interdependent sociotechnical theories do not understand technology as something static that is external to social contexts but as part of that context.[16 17] The implementation of information technology (IT) in healthcare settings has been explored from such sociotechnical perspectives.[16 17] IT does not operate in isolation from social and organisational contexts and as such plays a potential role in both adding to system complexity, and hence unsafe practice, and also contributing to the safety work of people within these systems.[1 18]

Extending this view, sociomateriality is a combination of the material properties of the technology with social processes. Sociomateriality considers that properties of the technology may be fixed, therefore allowing for differences in use to be assigned to such social processes and contexts.[19] Technology has been conceptualised as offering affordances in which the material properties of the technology, through interaction with human agents, allow for the possibilities for certain actions.[20–22] This contrasts to approaches that might focus narrowly and deterministically on the functions and capacities of the technology. In the concept of affordances, technology has the possibility of both shaping human action and being shaped by different social processes and practices.[20] New rules and conventions may evolve in a dynamic interaction between the technology, users and contexts that then changes social processes and practices.[19 23] In this way, technology is not a casual facilitator but affords the possibility of action as one element of a complex social process.[19–21]

## Medication safety technologies

The WHO's Third Global Patient Safety Challenge: Medication Without Harm places medication safety as a global priority.[24] Prescribing errors in general practice are an important and expensive cause of preventable safety incidents, illness, hospitalisations and deaths.[25] Medication safety across primary care operates within complex contexts that often involves collaboration between different general practice staff, between people and technology and is a fluid, negotiated, social process.[10 26 27] Prescribing errors have been considered to occur because of a multitude of error producing conditions including prescriber characteristics, patient characteristics, the working environment, the specific tasks and issues with technology.[28]

Prescribing safety in general practice is increasingly promoted through the use of specialist clinical decision support (CDS) tools used to provide clinicians with clinical knowledge and patient-related information to improve the quality of prescribing and reduce medication error.[29 30] The CDS system evaluated here has been in use across general practices in England and Wales since 2014. The CDS is embedded within the GP clinical system and provides a range of patient-specific messages that appear at the point of prescribing as a dialogue box or 'pop-up'. Prescribers have full discretion as to whether they accept the message. Previous evaluation of CDS systems focused on those designed to improve care outcomes often in hospital settings.[29 31 32] CDS has been seen as moderately successful, but it is widely acknowledged that prescribers often override alerts due to high volumes and limited perceived value ('alert fatigue'), and particularly if they interrupt clinical workflow by arriving at inappropriate times in the consultation.[32–35] 'Success' or 'failure' in implementation of IT has in the past been attributed to material properties of the technology and interoperability, user characteristics and organisational settings and processes.[36] It has however been suggested that discrete individual barriers and facilitators are less important than the complex interactions between those factors.[23 36] Engagement between the health professionals working in clinical settings and the technology may also inform understanding of the potential impact on changes in working practices, roles and responsibilities.[12 37] There has been little in the way of qualitative research that has utilised sociotechnical theory to understand how medication safety alerts might operate as part of the safety work undertaken by those working within primary care. This study aimed to apply sociotechnical theory to understand the use of CDS in the accomplishment of safety work in primary care, to understand how CDS transforms safety work and to understand how new forms of safety work are involved in the implementation and use of CDS.

## METHODS
### Study design
This qualitative study used semi-structured interviews with multiple stakeholders working as prescribers in general

practices in England. The study took place across nine different general practices within four separate clinical commissioning groups (CCGs), in the North West and East Midlands regions of England. The study was part of a wider qualitative longitudinal process evaluation of interventions designed to improve prescribing safety in primary care.[38 39]

## Sampling and recruitment of participants

The sampling was purposive and included people involved in prescribing medicines who had knowledge and experience of using CDS in their daily work in general practices within the four CCG areas, including GPs, general practice nurses and general practice-based pharmacists. Sampling and recruitment of general practices were facilitated through discussions with CCG managers. MJ approached potential practices between December 2017 and November 2019, by email or telephone and invited them to take part. In addition, MJ visited CCG meetings of groups of practices. After each practice had indicated that their staff might be interested in taking part, the practice was visited by MJ to further explain the study and provide written information. Practices consented to take part in the study before individual staff were approached. Individual general practice staff were approached directly by telephone or email, or through liaising with the general practice manager. A total of 41 practices were approached; nine practices consented to take part and 32 either declined to take part or did not respond. Practices commonly gave time and staffing commitments as reasons for not taking part. Each potential participant was provided with written information about the study and given a minimum of 24 hours to decide if they wished to take part. MJ then contacted those staff wanting to take part by telephone or email to arrange a convenient time for the interview. The interview process was two stage with follow-up interviews conducted approximately 12 months after the first interview with a convenience sample of different participants (n=7) in order to understand any changes that may have occurred and if any new themes were emerging.

## Data collection

The semi-structured interview schedule was developed by MJ and RNK drawing on previous studies in medication safety in primary care.[12 17 40] This was designed to illicit how CDS was used by the participants in the context of medication safety work (see online supplemental appendix S1). In order to ensure it remained appropriate the interview schedule was reviewed throughout data collection. Interviews continued alongside data analysis in an iterative approach until data saturation was reached. We understood data saturation as being achieved when we judged further interpretation of codes and themes would not provide further insights.[41] All interviews were conducted by MJ and took place face-to-face at the participants' usual place of work (general practice, n=19) or by telephone (n=4); all were digitally audio-recorded.

Participants were offered a £20 shopping voucher per interview as acknowledgement of their time. Twenty-three interviews were conducted with 20 participants (14 GPs, 3 nurse prescribers and 3 practice pharmacists). Follow-up interviews were conducted with 7 participants (5 GPs, 1 nurse, 1 practice pharmacist). The first interviews were conducted between February 2018 and November 2019 with follow-up interviews conducted between August 2019 and June 2020. Interviews ranged in duration from 13 to 41 min with a mean length of 31 min.

## Data analysis

Data analysis was informed by interdependent and constructionist sociotechnical theories and models.[14 15 23] MJ led the analysis, which was thematic and conducted alongside data collection following verbatim transcription. The analysis followed two parts: first, an inductive thematic approach informed by Braun and Clarke[42] and second, a deductive template approach which involved developing a coding template from the first stage.[43] For the initial inductive thematic approach, MJ used QSR NVivo 12 to organise the data and inductively coded early interviews to identify features, patterns, groups of codes and potential themes to build a preliminary framework for application to the data and to inform further data collection. These codes were discussed, in the context of sociotechnical theory, with the wider research team (RNK, N-ES, LL, AC, AShamsuddin). These discussions led to the development of a further coding template that was applied to the full data set, refined and reworked. From this coding, the final themes as presented in the results were interpreted.

## Patient and public involvement

AC is a patient and public representative who was involved in all aspects of the study and is a co-author of the paper.

## RESULTS

Interpretative data analysis found a complex network of relationships between the technology, different users and the contexts in which the technology was utilised. Safety work and the use of CDS were contextually situated in this network and reciprocally interrelated with it.

Three broad, interconnected themes were interpreted from the interview data.
► The use of CDS within organisational and social practices and workflows;
► The use of CDS within the interplay between prescribers, patients and populations;
► Sociomateriality—the affordances embedded in CDS.

## The use of CDS within organisational and social practices and workflows

For many prescribers, the use of CDS reflected their established commitment to promoting safety within their practice which was described as illustrative of their a 'safety climate'.

I think we are a safety conscious practice […] we don't want mistakes to happen so therefore creating a safe climate where you feel you are doing things safely is actually really important to us as a practice. **GP12**

Organisational culture varied. One GP reflected on these variations across different sites of the group practice in which they worked. One site had a strong focus on being 'traditionally good prescribers' who had 'work(ed) very hard […] on the prescribing (**GP3**). Such 'good' prescribing was seen to be about stability which (**GP3**) suggested 'creates much improved prescribing' in contrast to another site of their group practice which, they said, had high staff turnover. This was reflected in the number of CDS alerts seen. Having a 'safety system' was a way of avoiding many alerts. A safety culture that included medication reviews and which had been 'done …for so long that it's relatively slick' was perceived as conducive to utilising alerts from CDS systems in a way that avoided being overwhelmed.

> Medication reviews I do here. This probably might explain why we've got less (alerts) here. We do a six month or annual review on people that are only on two things here, and we've done it for so long that it's relatively slick. Whereas other places where they've not had review I'm going into a jungle almost of interactions and possible problems. They've never had any safety system before, so there's lots of alerts will go on. **GP3**

Organisational practices involved workflows where the use of the system was most pertinent in medication reviews. The following GP utilised CDS mostly when undertaking repeat prescribing. This was perceived to lead to disregarding alerts because 'the patient's not in front of you' and because of the volume of prescriptions that needed repeat authorisation.

> I don't often use it (CDS support) in consultations, because I suppose that's one of the problems of it, […] the time it crops up in workflows most, is when you're signing today's 60 repeat prescriptions, so the patient's not in front of you, but there's a medication review overdue, […] (the alert) flashes up and tells you that something minor could need changing, and you sort of let it go. **GP13**

## The use of CDS alerts within the interplay between prescribers, patients and populations

The characteristics of patients and roles of prescribers impacted on the CDS use for medication safety. Users with differing roles in general practice might not use CDS in the same way. The wider demographics of patient groups and broader holistic needs of individual patients were considered relevant to how prescribing would be prioritised and how CDS was used. For one GP, changes to specific individual medications, as alerted by CDS, were not patient's primary concern.

> …if you've got a patient with 20 different drugs, […] they've just lost their job, they're about to become homeless, they've got epilepsy, depression, anxiety, diabetes that's out of control, they've got 15 appointments, but they've missed the last ten, and they've got a safeguarding issue with their child […] There are much, much bigger issues for them. **GP5**

Individual patient needs, particularly for patients with complex medical problems or multi-morbidity, could impact on the ways in which CDS alerts were responded to by prescribers. One GP suggested that it was difficult to respond to alerts if patients were 'on long lists of medication already'. **GP14**

Alerts were rejected if the prescriber disagreed and were considered inappropriate to the patient because of other morbidity, frailty or their social situation. Safety work was accomplished by the prescribers balancing alerts against their own knowledge and understanding of individual patient circumstances. In this specific case, leaving a patient with epilepsy without medication as the alert had suggested was perceived as less safe.

> Sometimes it's hazardous not to prescribe […] some of the epilepsy drugs you can't give them safely because they're dangerous drugs; and it's telling you all these warnings, but you've got to give something because it would be more dangerous to leave somebody fitting […] somebody with bad COPD (chronic obstructive pulmonary disease) and they're on a host of other drugs, there really isn't an antibiotic that you can give that doesn't have a risk of cardiac arrhythmia or something like that; (but), I can't leave them without the antibiotic […] we've got to give something. **GP1**

Safety work was also accomplished by different prescribers within the context of responsibilities, and moral and ethical behaviour. Prescribers reflected how alerts did not remove the responsibility of a prescriber's signature and prescribers still had to justify their actions.

> I don't think anyone tool […], you shouldn't rely on just one thing at all. You are still responsible for prescribing and what you prescribe so you've still got to justify it. You can overrule … **GP5**

Alerts were also seen within the context of balancing different responsibilities towards the patient, the wider practice population and the management of resources.

> So our prime responsibility is to the patient who's in front of us now. But, at the same time, we have a responsibility to our whole practice population, and to the country, of managing resources correctly. So, we have three different responsibilities that exist, almost like a holy trinity, they're all there at the same time […] And so anything that allows you to focus on one, while monitoring your progress on the other, is likely to be helpful, or feels helpful. So, you know, if I have someone in front of me who needs an antidepressant,

and the current evidence is that sertraline is better than citalopram, but I didn't get to see that, if (CDS) comes up and says, actually we're now supposed to be giving sertraline, I think, oh that's great, 'cause that's a helpful thing, 'cause it helps me monitor my other obligations at the same time. **GP18**

Knowing that there was patient information within alerts meant that the clinician had a moral obligation to act on that information.

'….there's a degree of responsibility, if you know that you have a tool and, you know, and there's potentially a person sat there, who could have been looked at, who hasn't been looked at, and is subject to potential harm. You've got responsibility, haven't you, as a clinician, because that information is there for you to access'. **GP15**

Safety work and prescribing was undertaken differently by different prescribers. This nurse considered that while they had been prescribing for 11 years it was generally felt among nursing staff that prescribing was an 'extended role'. This led to a lack of confidence, a fear of making mistakes and the seeking of support.

I think the way nurses think because we see it as an extended role, it's not part of this. I think there's that element of it and we're always worried, we're always covering our back all the time. A doctor said to me once that it's the mind set of how we're trained, nurses are trained to question, have I done something wrong […] it's how we think, scared to do something wrong. **Nurse 1**

In contrast, the use of CDS was seen as being dependent on experience and expertise. This GP suggested that it was easy to use and decide whether to accept alerts or not because they had many years' experience and was the prescribing lead for the practice. They suggested that they were more likely to rely on their own experience and knowledge than less experienced prescribers who might be more likely to rely on alerts.

Having been a GP for a long time, I'm also the prescribing lead for the practice, […] obviously, that relies on your own experience, skills, confidence, that sort of thing. I think probably as maybe one of the relatively older GPs, […] having a reasonable amount of experience and confidence in what you're doing, probably means that you cancel it off more often than a younger one. **GP14**

For this GP, there was a clear contrast between how they undertook safety work using CDS and how that was conducted by their practice pharmacist. Practice pharmacists reported that they saw patients for longer appointment times and would be more likely to be dealing with medication reviews for patients with long-term conditions. Their contrasting work practices were related to differences in response to CDS alerts with the GP less likely to

be methodical and more likely to not change medications in case the patient complained.

…it's been interesting seeing how our pharmacist works, now that (they are) seeing patients and doing (their) own medication reviews. (They are) much more thorough and […] will challenge people a lot more and say, look you've been on this thing, I'm not sure you necessarily need that, or you may be need something slightly different, and will go through it in a more methodical fashion. I think we do probably in our consultations, sort of, go through things a little bit more quickly. **GP4**

### Sociomateriality—the affordances embedded in the CDS

CDS was used within electronic health record (EHR) systems and was dynamically linked to other databases and clinical systems that could provide alerts. As a consequence, this impacted on how CDS was used, how users interacted with it and how the alerts were responded to. Part of this was associated with a frustration about what different technologies could or could not do, or what could or could not be done with different technologies. Prescribers talked of CDS when embedded with one particular clinical system 'failing to launch' …'it's just so many more clicks on particular things of your screen' **GP12** or 'the number of warnings that comes up in (name of clinical system) is far too many' **GP13.** This was said to mean 'it slows down and frustrates you as you're doing an important task but at the same time you're not getting the advice from it as well.' **GP13** Affordance in this way was not just about non-use or abandonment but about how that made users perceive the system and feel about using it, which then in turn led to a negative response. Such frustration with the CDS could lead to important medication safety issues.

We had a significant event where a medication was prescribed to a patient that shouldn't have been. We know that at least one of the factors in that was the fact that in the set up process of (the clinical system) different doctors chose different security settings for medicines safety… **GP13**

The perception that CDS was not providing accurate and useful alerts could mean that the system was ignored and therefore safety work potentially compromised.

In contrast, some participants described how CDS provided affordances for prescribers to undertake safety work. This GP considered the alerts not intrusive, easily overridden and providing useful and sensible safety advice. Such safety advice was seen as important since it could mitigate against safety problems for patients in the future.

I don't find them too intrusive really. I mean it is another thing that happens that didn't happen before but it's as we discussed before, it was they are relatively easy to override and the suggestions that are being

made generally feel quite sensible […] And if it's a safety thing, then generally I've been quite pleased that it's happened because it's given me an opportunity to make a change before potentially something that might cause me a problem later down the line. **GP4 (Follow-up interview)**

CDS was seen as providing prompts to safer prescribing that challenged prescribers to think about what they were about to prescribe. This could take the form, as described by this GP of avoiding a prescribing error.

… today it's prompted me not to prescribe the wrong dose of oxycodone. So off a picking list I was close to picking the wrong strengths of oxycodone. And that came up with an (CDS) prompt which was great. […], so I was very happy to have something held up in front of my face going, are you sure about that? **GP4**

## DISCUSSION

The use of CDS for medication safety in primary care was seen to be contextually situated in the social practices of safety work. This builds on understandings of healthcare settings as complex systems with different characteristics including people, tasks and technology.[3] In this study, safety work was accomplished within safety cultures, which varied for patients and prescribers and therefore shaped their particular forms of engagement with the CDS and safety work. The use of sociotechnical theory here extends current thinking in patient safety particularly in the ways that safety work was co-constituted with the use of CDS alerts.

### Safety work in primary care

The safety work undertaken by prescribers when using CDS was often shaped by the perceived health needs and broader characteristics of the patient. Where CDS support was less easy to accommodate to the holistic practices of prescribers, in terms of their relationship with patients, it was typically underutilised. Specifically, for patients with multiple and complex care needs, prescribers suggested it was necessary to consider factors other than prescribing, and relatedly the existing CDS system presented functional barriers to working with patients on multiple medicines. A recent systematic review and meta-analysis of medication harm across health settings found the highest incidence of medication harm in patients with high comorbidity and related polypharmacy.[44] It has previously been suggested that decision-making is made more complicated for clinicians when dealing with patients with multi-morbidity.[45 46] As such, the underutilisation of CDS might be seen as stemming from a lack of appreciation among designers and policymakers about how prescribing safety is shaped by the particular relationship between prescriber and patient, in the context

of the former's prevailing safety practices and the latter's particular health needs.

We found that the utilisation of CDS for safety work within primary care was linked to the affordances within the technology and significantly how these related to the perceived needs of the patient and the associated approach to medical decision-making. That is, GP's safety work often involved ignoring alerts where they were perceived as inaccurate, overwhelming or irrelevant to the complexity of the decision-making for a given patient. Although our study did find that in other relational contexts, that is, with less complex patients, or where there was more time available such as in consultations with practice pharmacists, CDS did provide prescribers with valued and useful alerts that were seen as enhancing their safety work.

CDS alerts were not neutral, in that they provided a specific recommendation to the prescriber. This could be seen as form of 'digital nudging' whereby technology might be seen to influence choices and 'nudge' people into specific behaviours.[47] That the prescribers here did reject alerts suggests that this was an active rather than passive response to the alert. CDS alerts afforded the possibility of making prescribing changes but were only acted on if the prescriber considered it appropriate within the given contexts of their practice. The system provided prompts and challenges in the form of alerts, which could be accepted, and acted on in different ways, or ignored and thus there was a recursive relationship between people and the technology.[14] Responding to CDS alerts was therefore not a linear and binary process but one that was fluid and contextually bound. Prescribers' responses to safety alerts were based on their expertise and knowledge. Previously, Swinglehurst et al[10] found that safety work in repeat prescribing was embedded in different types of organisational structures. This study reinforces the sense that primary care and medication safety operate as complex systems with multiple and interconnected parts.[1] We found that the organisational culture and what prescribers referred to as a 'safety climate' or 'safety culture' of particular importance with organisational practices such as workflows and tasks shaping the use of CDS. This reinforces that safety work is a social process operating within multidimensional and dynamic systems.[8 9 48] As well as organisational processes shaping the use of CDS, CDS alerts shaped the way prescribers accomplished safety work as they variously adapted their clinical practices and responded differently to them.[1 3] This included prescribers reflecting on their moral and ethical responsibilities towards their patients and the wider community. This further highlights how safety work requires a flexible approach that is adaptive and responsive to different contextual situations and does not over rely on technology.[10] Our study builds on understandings of alert fatigue in that underutilisation cannot be therefore simply seen in terms of the volume and relevant of alerts but that alerts are part of the social process of managing safety within complex work in primary care.

## STRENGTHS AND LIMITATIONS

A particular strength of this study is the focus on sociotechnical theory, which enabled a nuanced understanding of the ways in which the CDS was utilised. By using this theoretical lens, we could further understand how the CDS alerts and those already embedded in the EHR were utilised within the context of organisational culture and prescribers workflows. This builds on the sociotechnical theory particularly that around the concept of affordances within technology that enable human actions.[20 21] Additionally, a further strength was the longitudinal design with a variety of stakeholders from different professions interviewed over time across multiple geographical locations. A full range of different prescribers in different roles (GPs, nurses and pharmacists) were interviewed but there could have been greater breadth across general practices. The present study found some variation between different prescribers (pharmacists, GPs or nurses) in the ways they adopted the CDS for safety work, but we could only explore this in a limited way given our study sample. Further understanding how safety work is accomplished in the context of different users of the technology in future research would be useful. Many practice staff were also drawn from two of the four targeted CCG areas due to recruitment difficulties related to GP time and workload. Furthermore, interviewing patients to gain their perspective may well have been useful. This study also relied solely on interview data and may have benefited from observations or other ethnographic approaches that might have uncovered 'work-as-done'.[5]

### Implications for future research, policy and practice

CDS systems for medication safety have been utilised in primary care with varying success.[31 34 49] Since CDS systems operate within other technology such as the EHR, it may be important to understand how different technologies interact and offer different ways people can engage in safety activities. Understanding open systems and the complexity in healthcare has been highlighted as important in healthcare evaluation.[1 50] Investment into CDS systems needs to be judged against how those systems will be used not just from a functionality aspect but also from the view of who the users will be and for which individual and groups of patients they will be utilising the system for. How CDS are utilised for patients with multi-morbidity and polypharmacy will be an important consideration here. An important finding of our study was that functionality was linked to patient characteristics and prescriber motivations, in that where multiple alerts were received owing to patients having polypharmacy the system was then underutilised and the alerts could be ignored. Since this group of patients is at most risk from a prescribing error, this has important implications for practice. Multidisciplinary reviews of patients with polypharmacy have been found useful in other contexts including care homes.[51 52]

There is a recursive interrelationship between CDS alerts and safety culture in that CDS can enhance safety culture but may also be utilised differently within existing safety cultures as prescribers adapt their practices.[1 3] Further research could valuably explore how technology to enhance medication safety can shape organisational safety work and not just be utilised within a safety climate but be instrumental in building that safety culture. Understanding how alerts can adapt that safety culture and how prescribers may adapt to the alerts to form new emerging safety cultures is a challenge that will help improve the efficacy of CDS systems, enhance prescribing safety and help to further understand how safety work is achieved in primary care.

## CONCLUSIONS

Sociotechnical theory enabled us to understand how the use of CDS for medication safety in primary care is context bound and dependent on a complex and reciprocal network of relationships between the technology, different users and the places and spaces in which the technology was utilised. Where prescribers perceived their practices to have a safety culture, this could impact on the utilisation of CDS. The use of CDS for medication safety work was dependent on who responded to alerts and for which patients the alerts were received. Since alerts for patients with multimorbidity and polypharmacy were reported to be underutilised, an important implication and strategy of our findings is how prescribers may adapt their practices to ensure medication safety for this group of patients. Prescribers adaptation of their practices in response to alerts has implications for the use of decision support to ensure that the contexts into which it is implemented and therefore the ways in which safety work is accomplished are taken into account.

**Author affiliations**

[1]NIHR Greater Manchester Primary Care Patient Safety Translational Research Centre, The University of Manchester, Manchester, UK
[2]Centre for Pharmacoepidemiology and Drug Safety, Division of Pharmacy and Optometry, School of Health Sciences, The University of Manchester, Manchester, UK
[3]Division of Primary Care, School of Medicine, University of Nottingham, Nottingham, UK
[4]School of Health Sciences, University of Hull, Hull, UK
[5]Division of Community Health Sciences, University of Edinburgh, Edinburgh, UK
[6]School of Social Policy, Health Services Management Centre, University of Birmingham, Birmingham, UK
[7]Suicide, Risk and Safety Research Unit, Greater Manchester Mental Health NHS Foundation Trust, Manchester, UK

**Contributors** TA secured funding. MJ, RNK, TA and JW advanced the idea for the study. RNK and TA provided study management. MJ collected the data. MJ led on data analysis with contributions from N-ES, LL, AC, RNK, ASheikh and AShamsuddin. MJ drafted the manuscript which was reviewed and commented upon by all authors. All authors approved the final manuscript. MJ is responsible for overall content as guarantor.

**Funding** This study was part of the NIHR PRoTeCT Programme grant. PRoTeCT is funded by the NIHR Programme Grants for Applied Research Programme (1214-20012). Additionally, MJ, RNK, TA, LL and N-ES received funding from the National Institute for Health Research through the Greater Manchester Patient Safety

Translational Research Centre (NIHR Greater Manchester PSTRC) grant number PSTRC-2016-003.

**Disclaimer** The views expressed are those of the author(s) and not necessarily those of the NIHR or the Department of Health and Social Care.

**Competing interests** None declared.

**Patient and public involvement** Patients and/or the public were involved in the design, or conduct, or reporting, or dissemination plans of this research. Refer to the Methods section for further details.

**Patient consent for publication** Not applicable.

**Ethics approval** This study involves human participants and ethical approval was granted by the University of Manchester Research Ethics Committee (Ref: 2017-0842-3977). Further approval to conduct the research was granted by the NHS Health Research Authority (IRAS project ID: 233079) and by the research and development offices for the NHS participating regions. Participants gave informed consent to participate in the study before taking part.

**Provenance and peer review** Not commissioned; externally peer reviewed.

**Data availability statement** No data are available. This is a qualitative study confined to relatively small groups of health care professionals in specific roles, particularly GP staff. Making the full transcripts publicly available could therefore potentially lead to the identification of participants. Our ethics approval was granted based on the anonymity of the individuals consenting to participate and specifically referred to only anonymised quotations being used in reports. As such the participants did not consent to full their transcript being made publicly available. Therefore, data cannot be made publicly available.

**ORCID iD**
Mark Jeffries http://orcid.org/0000-0002-6882-0350

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
