## [Reviewer comments · BMJ Open]

ARTICLE DETAILS

TITLE (PROVISIONAL)	Using sociotechnical theory to understand medication safety work in primary care and prescribers' use of clinical decision support: a qualitative study
AUTHORS	Jeffries, Mark; Salema, Nde-Eshimuni; Laing, Libby; Shamsuddin, Azwa; Sheikh, Aziz; Avery, Tony; Chuter, Antony; Waring, Justin; Keers, Richard

VERSION 1 – REVIEW

REVIEWER	Petrakaki, Dimitra University of Sussex, Department of Management
REVIEW RETURNED	09-Nov-2022

GENERAL COMMENTS	Using Sociotechnical Theory to Understand Medication Safety Work in Primary Care and Prescribers use of Clinical Decision Support: A Qualitative Study. The paper aims to understand the relationship between safety work and use of clinical decision support systems in the context of prescribing in primary care. It draws upon a qualitative study and interview data with different types of prescribers, GPs, nurse prescribers and GP-based pharmacists. The paper contributes to the literature by exploring how alerts enabled by systems play out with the context within which systems are used and its safety culture and illustrates the complexity as clinical decision making interacts with the system's alerts and with other parameters such as prescriber's work experience and patients' conditions. Theoretically, the paper draws upon a sociotechnical perspective to explore the role of technology as both an enabler and a barrier to safe practice, bringing to the fore the complexity of the question the authors aim to address. I provide below some questions/comments for the authors to consider when they revise their paper. The paper would become clearer if it provides some examples to illustrate what safety work entails. The literature on sociotechnical studies is rather vast and I would like to invite the authors to position their paper to a more specific perspective within this field. They refer to sociomateriality and to a theory of technological but do not specify which of these theories has primarily shaped the analysis of this paper and wh. The findings are interesting in terms of illustrating how alerts play out with clinicians' decision making. The paper could potentially contribute to the emerging field of digital nudging as a way of shaping behaviour, in this case safety work. Your findings also point to several ethical questions that prescribers encounter and
--

	show how their decisions are influenced by a complex interplay between alerts, ethics and professionalism/professional identity. Do you have any findings to illustrate this? Could you please discuss how much discretion prescribers could use when they encountered an alert and offer an additional description of the functionality/affordances of the system and of its use by users? In the analysis, the paper seems to use 'functionality' and 'affordances' interchangeably. Could you please define the two concepts and how they differ with each other? I was wondering how consistent your concluding observations are across the different prescriber roles. Overall, I think this study is timely, draws upon and promotes a sociotechnical understanding to these complex digital health initiatives and offers some rich and fascinating data. I enjoyed reading your work thoroughly!
--	---

REVIEWER	Garfield, Sara University College London, School of Pharmacy
REVIEW RETURNED	04-Jan-2023

GENERAL COMMENTS	This is a very interesting and relevant study. I have a few suggestions. 1. Methods: What were the follow up interviews? Was each healthcare professional interviewed twice? Discussion: Can you add more specifically how your findings have added to what we already know about alert override? Conclusion: Can you make this more specific as to the implications of your findings.
--

VERSION 1 – AUTHOR RESPONSE

Reviewer: 1 Comments		
The paper aims to understand the relationship between safety work and use of clinical decision support systems in the context of prescribing in primary care. It draws upon a qualitative study and interview data with different types of prescribers, GPs, nurse prescribers and GP-based pharmacists. The paper contributes to the literature by exploring how alerts enabled by systems play out with the context within which systems are used and its safety culture and illustrates the complexity as clinical decision making interacts with the system's alerts and with other parameters such as	Thank you for taking the time to review our paper.	

prescriber's work experience and patients' conditions. Theoretically, the paper draws upon a sociotechnical perspective to explore the role of technology as both an enabler and a barrier to safe practice, bringing to the fore the complexity of the question the authors aim to address. I provide below some questions/comments for the authors to consider when they revise their paper.		
1. The paper would become clearer if it provides some examples to illustrate what safety work entails.	Thank you for this helpful comment . We have added the following at the end of the first paragraph of the Introduction. This includes citations for two additional references Jeffries (2020) and Grant (2016). This reads: “Safety work has been seen to entail an understanding of safety culture amongst staff⁹, systems and collaboration between reception, administration and medical staff¹⁰, articulation work¹¹ development of relationships and engagement with stakeholders^{12 13} and management of workload.⁸”	Introduction , P6
2. The literature on sociotechnical studies is rather vast and I would like to invite the authors to position their paper to a more specific perspective within this field. They refer to sociomateriality and to a theory of technological but do not specify which of these theories has primarily shaped the analysis of this paper and wh.	Thank you for this helpful point. Our perspective would be one that draws upon interdependent and constructionist theories and approaches. In this we understand technological use to be a non-linear social process involving an interdependent interplay between organisational contexts , human actors and technological infrastructure. We draw upon the work Greenhalgh et al (2014, 2016), and Greenhalgh and Stones (2010). We also consider the ways technology might be utilised from , as you say, a sociomateriality perspective wherein technology is seen as offering a range of possibilities within social and organisational contexts. We see these two perspectives as complimentary in framing our understanding of our CDS maybe utilised in primary care. We have made some changes to the second paragraph of the introduction to reflect this and added an additional reference (Greenhalgh et al., 2016). This now reads:	Introduction , P6-7

	Sociotechnical theory, that draws upon interdependent and constructionist theories, considers that people and technology are reciprocally and recursively related and as a consequence the outcomes of the relationships between the social, human agency and technology are considered as interdependent and not simply as the interactions between homogenous unique elements. ^{14 15} This approach focuses upon the social processes involved in the use of the technology since it sees technological use and adoption as a social practice that may involve negotiation and conflict. Interdependent sociotechnical theories do not understand technology as something static that is external to social contexts but as part of that context. ^{16, 17.} The implementation of Information Technology (IT) in health care settings has been explored from socio-technical perspectives that consider not only the ways technology is used and user characteristics, but the social and organisational context into which technology is embedded. ^{18 19} IT does not operate in isolation from social and organisational contexts and as such plays a potential role in both adding to system complexity and hence unsafe practice, and also contributing to the safety work of people within these systems. ^{1 20} We have also clarified this with a sentence at the beginning of the data analysis section of the methods where we also add citations of three papers: Orlikowski and Scott 2008, Greenhalgh 2016, and Klecun 2016. This sentence reads: “Data analysis was informed by interdependent and constructionist sociotechnical theories and models. ^{14 – 16”}	Methods: Data analysis, P11
3. The findings are interesting in terms of illustrating how alerts play out with clinicians’ decision making. The paper could potentially contribute to the emerging field of digital nudging as a way of shaping behaviour, in this case safety work. Your findings also point to several	Thank you for these helpful observation points. We have added a section to the findings that reflects how responsibility and ethical behaviour was indeed part of the way prescribers had to balance decisions around safety work.	Results, P14-15

ethical questions that prescribers encounter and show how their decisions are influenced by a complex interplay between alerts, ethics and professionalism/professional identity. Do you have any findings to illustrate this?	We have added this text to the Results: Safety work was also accomplished by different prescribers within the context of their responsibilities, and moral and ethical behaviour. Prescribers reflected how alerts did not remove the responsibility of a prescribers' signature and prescribers still had to justify their actions. I don't think any one tool [...], you shouldn't rely on just one thing at all. You are still responsible for prescribing and what you prescribe so you've still got to justify it. You can overrule ... GP5 Alerts were also seen within the context of balancing different responsibilities towards the patient, the wider practice population and the management of resources. ...so our prime responsibility is to the patient who's in front of us now. But, at the same time, we have a responsibility to our whole practice population, and to the country, of managing resources correctly. So, we have three different responsibilities that exist, almost like a holy trinity, they're all there at the same time [...] And so anything that allows you to focus on one, while monitoring your progress on the other, is likely to be helpful, or feels helpful. So, you know, if I have someone in front of me who needs an antidepressant, and the current evidence is that sertraline is better than citalopram but I didn't get to see that, if (CDS) comes up and says, actually we're now supposed to be giving sertraline, I think, oh that's great, 'cause that's a helpful thing, 'cause it helps me monitor my other obligations at the same time. GP18 Knowing that there was patient information within alerts meant that the clinician had a moral obligation to act upon that information.	
---	--	--

	....there's a degree of responsibility, if you know that you have a tool and, you know, and there's potentially a person sat there, who could have been looked at, who hasn't been looked at, and is subject to potential harm. You've got responsibility, haven't you, as a clinician, because that information is there for you to access. GP15 We have also reflected upon the issues you have raised in the discussion. We agree that digital nudging is pertinent here and have added some comments into those reflections and provided a new citation to the work of Weinmann et al., 2016. At the beginning of the first paragraph on page 20 we have added he following. CDS alerts were not neutral in that they provided a specific recommendation to the prescriber. This could be seen as a form of 'digital nudging' whereby technology might be seen to influence choices and 'nudge' people into specific behaviours.⁴⁷ That the prescribers here did report rejecting alerts suggests that this was an active rather than passive response to the alert. Towards the end of this paragraph we have added this sentence: This included prescribers reflecting upon their moral and ethical responsibilities towards their patients and the wider community.	Discussion, P20 Discussion, P21
4. Could you please discuss how much discretion prescribers could use when they encountered an alert and offer an additional description of the	Prescribers had full discretion in their acceptance of alerts. The CDS system was embedded into the GP electronic health record and provided a suite of alerts some of which were related to cost saving, some best practice	Introduction , P8

functionality/affordances of the system and of its use by users?	and safety. We have added to the Introduction to provide this further context. We have added this at page 8: The CDS system evaluated here has been in use across general practices in England and Wales since 2014. The CDS is embedded within the GP clinical system and provides a range of patient specific messages that appear at the point of prescribing as a dialogue box or 'pop-up'. Prescribers have full discretion as to whether they accept the message.	
5. In the analysis, the paper seems to use 'functionality' and 'affordances' interchangeably. Could you please define the two concepts and how they differ with each other?	Thank you for this helpful point. We would see affordances as the opportunities (or constraints) for action through interaction, and functionality as the ability to perform a set task through the use of technology. We also draw upon socio-materiality which offers the scope for fixed properties of technology and allows for the focus upon context and social process as we have described in our response to your point above. We understand the tension here between affordances that are part of social practices and technology in use, and functionality that might be considered a deterministic approach where capacities residing within the technology determine whether and how it is used. Given what we say in the introduction (particularly in the strengthened version in response to your point above) that we see technology as not operating in isolation but reciprocally and recursively related to people, and therefore a social practice, we think it probably better to use affordances throughout rather than functionality. We think this places our manuscript more clearly within the sociotechnical theoretical approaches we have outlined in it and above to reflect this decision. We have made a number of changes to the manuscript as a consequence. In the introduction the end of paragraph 2 on page 7 now reads: This contrasts to approaches that might focus narrowly and deterministically upon the functions and capacities of the technology. In the concept of affordances , technology has	Introduction , P7

	the possibility of both shaping human action and being shaped by different social processes and practices. ²¹ New rules and conventions may evolve in a dynamic interaction between the technology, users and contexts that then changes social processes and practices. ^{16 17} In this way technology is not a casual facilitator but affords the possibility of action as one element of a complex social process. ^{17, 21, 22} Within the results section we have changed the paragraph on page 17 to now read: Sociomateriality – the affordances embedded in the CDS. CDS was used within Electronic Health Record (EHR) systems and as dynamically linked to other databases and clinical systems that could provide alerts. As a consequence this impacted upon how CDS was used, how users interacted with it, and how the alerts were responded to. Part of this was associated with a frustration about what different technologies could or could not do, or what could or could not be done with different technologies. Prescribers talked of CDS when embedded with one particular clinical system ‘failing to launch’ ... ‘it’s just so many more clicks on particular things of your screen’ GP12 or ‘the number of warnings that comes up in (name of clinical system) is far too many’ GP13. This was said to mean ‘it slows down and frustrates you as you’re doing an important task but at the same time you’re not getting the advice from it as well.’ GP13. Affordance in this way was not just about non-use or abandonment but about how that made users perceive the system and feel about using it, which then in turn led to a negative response. Such frustration with the CDS could lead to important medication safety issues. We have made a few minor changes elsewhere as highlighted in the manuscript.	Results, P17
--	--	-------------------------

6. I was wondering how consistent your concluding observations are across the different prescriber roles.	We have highlighted some differences between prescribers in the results section on page 13. However we agree that this needs commenting upon, is potentially a limitation of our study and have therefore added the following to the limitations section of the discussion. The present study found some variation between different prescribers (Pharmacists, GPs or Nurses) in the ways they adopted the CDS for safety work but we could only explore this in a limited way given our study sample. Further understanding how safety work is accomplished in the context of different users of the technology in future research would be useful.	Results, P16 Discussion, Strengths and limitations, P21-22
Reviewer 2		
1. Methods: What were the follow up interviews? Was each healthcare professional interviewed twice?	Follow up interviews were not conducted with all participants but with a convenience sample of 7 different participants. This is stated on page 8 at the end of the sampling and recruitment section of the Methods. For further clarification we have added the following to the section on data collection on page 9: "Follow-up interviews were conducted with 7 participants (5 GPs, 1 nurse, 1 practice pharmacist)."	Methods-Data collection. P9
2. Discussion: Can you add more specifically how your findings have added to what we already know about alert override?	Our aims with this paper were to explore how CDS was used in the accomplishment of safety work in primary care. Our focus was therefore not specifically drawn to the overriding of the alerts but more about the recursive interrelationship between CDS alerts and safety culture. However your point is valuable and important since, as we suggest alerts require adaptation in work by prescribers. We feel that we have highlighted in the discussion that the overriding or underutilisation of alerts was specifically related to patient needs, prescriber characteristics and the complex contexts in which medical decision making is enacted in primary care. We respectfully draw your attention to the third paragraph of the discussion on page 20. We feel that our paper adds to an understanding of alert override in	Discussion, P20, P21

	that it places that within the recursive relationship between the prescribers and the technology and moves beyond linear cause and effect explanations of such underutilisation to understandings that as fluid and contextually bound. For clarity we have added the following to the discussion. Our study builds upon understandings of alert fatigue in that underutilisation cannot be therefore simply seen in terms of the volume and relevant of alerts but that alerts are part of the social process of managing safety within complex work in primary care.	
3. Conclusion: Can you make this more specific as to the implications of your findings?	Thank you for this helpful point. We have made some changes to the conclusion as a result that are described below. Sociotechnical theory enabled us to understand how the use of CDS for medication safety in primary care is context bound and dependent upon a complex and reciprocal network of relationships between the technology, different users and the places and spaces in which the technology was utilised. Where prescribers perceived their practices to have a safety culture this could impact upon the utilisation of CDS. The use of CDS for medication safety work was dependent upon who responded to alerts and for which patients the alerts were received. Since alerts for patients with multimorbidity and polypharmacy were reported to be underutilised an important implication and strategy of our findings is how prescribers may adapt their practices to ensure medication safety for this group of patients. Prescribers' adaptation of their practices in response to alerts has implications for the use of decision support to ensure that the contexts into which it is implemented and therefore the ways in which safety work is accomplished are taken into account.	Conclusion, P23

VERSION 2 – REVIEW

REVIEWER	Petrakaki, Dimitra University of Sussex, Department of Management
REVIEW RETURNED	28-Mar-2023

GENERAL COMMENTS	Thank you for submitting your revised paper, which I have read with great interest. The authors have managed to engage with all comments and the paper is now much stronger theoretically and with clearer contributions. The literature review sets up the sociotechnical concepts nicely and provides a connection between IT affordances and prescribing errors. It was great to see the enriched section of your findings whereby you discuss how these prompts play out with prescribers' responsibility and with a range of perimeters, factors and dilemmas prescribers have to consider in their daily work life. I enjoyed the links you made to digital nudges and alert fatigue. This is a really interesting study in the context of electronic prescribing and safety work.
--

VERSION 2 – AUTHOR RESPONSE

Reviewer: 1 Comments		
Thank you for submitting your revised paper, which I have read with great interest. The authors have managed to engage with all comments and the paper is now much stronger theoretically and with clearer contributions. The literature review sets up the sociotechnical concepts nicely and provides a connection between IT affordances and prescribing errors. It was great to see the enriched section of your findings whereby you discuss how these prompts play out with prescribers' responsibility and with a range of perimeters, factors and dilemmas prescribers have to consider in their daily work life. I enjoyed the links you made to digital nudges and alert fatigue. This is a really interesting study in the context of electronic prescribing and safety work.	Thank you for taking the time to review our paper again and your kind remarks here.